# A Game Theoretic Model of Choosing a Valuable Good via a Short List Heuristic

**David M. Ramsey** 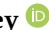

Faculty of Computer Science and Management, Wroclaw University of Science and Technology,
50-370 Wrocław, Poland; david.ramsey@pwr.edu.pl

**Abstract:** The Internet gives access to a huge amount of data at the click of a mouse. This is very helpful when consumers are making decisions about which product to buy. However, the final decision to purchase is still generally made by humans who have limited memory and perception. The short list heuristic is often used when there are many offers on the market. Searchers first find information about offers via the Internet and on this basis choose a relatively small number of offers to view in real life. Although such rules are often used in practice, little research has been carried out on determining, for example, what the size of the short list should be depending on the parameters of the problem or modelling how the short list heuristic can be implemented when there are multiple decision makers. This article presents a game theoretic model of such a search procedure with two players. These two players can be interpreted, for example, as a couple searching for a flat or a second-hand car. The model indicates that under such a search procedure the roles of searchers should only be divided when the preferences of the players are coherent or there is a high level of goodwill between them. In other cases, dividing the roles leads to a high level of conflict.

**Keywords:** game theory; job search problem; heuristics; short list; consumer behaviour

---

## 1. Introduction

The Internet contains a huge amount of information about offers to consumers. On the one hand, this wealth of offers makes it very easy to make purchases. However, when making very important purchases (e.g., a new flat or a second-hand car), final purchase decisions are made by humans with limited memory and abilities of perception. In such situations, decision makers (DMs) use search heuristics which maximise the probability of making a successful purchase over a wide range of possible scenarios, while keeping search costs relatively low.

This paper considers a game theoretic model of a pair of DMs searching for a valuable resource using an approach based on the short list heuristic. In the first stage of search, one of the DMs observes advertisements giving basic information regarding a large number of offers. On the basis of these advertisements, a short list of offers are chosen for closer inspection. The second DM then chooses the offer to be purchased on the basis of all the information gathered regarding each offer. The first stage of this process can be facilitated by the Internet. It is assumed that only the first DM incurs search costs in the first round, while both DMs incur costs in the second round of search. It is implicit in this assumption that the costs involved in the Internet search simply involve the time spent by the first DM in forming the initial short list. The costs in the second round involve the travel costs involved in observing the offers physically and both DMs observe the offers on the short list (or at least incur these travel costs), although the second DM makes the final decision. The second assumption may be relaxed when appropriate, but is adopted here to enable easier comparison with search procedures involving one DM. Due to the nature of the search processes that are considered here, it is assumed that the search costs incurred in the first round are much lower than those incurred in the second round.

Even when it might be theoretically possible to fully compare a set of offers based on the information immediately available, in practice it may be optimal to restrict search to a subset of the offers available (see Caplin et al. [1] and Ellis [2]). For example, the cognitive and/or time costs involved in comparing a large set of offers might be larger than the benefits that can be gained from exhaustive search of all the options.

In decision making procedures, the criteria taken into account and the time available are crucial factors in determining the success of DMs. Speier-Pero [3] presents results of an experiment where participants were asked to form short lists of candidates for job positions on the basis of data which were presented in various forms. Participants who had aggregated data and limited time were more successful at solving such problems than those who were given full data, but more time. Carvalho and Silverman [4] investigated how participants in an experiment chose investment portfolios. Choosing a portfolio based on simplified data gave a lower average gain, but also a lower level of risk. In addition, participants with a higher level of analytical skills were relatively successful when choosing portfolios based on non-aggregated data. It is thus clear that DMs are subject to limitations on their abilities of perception and decision-making. Giarlotta [5] gives an overview of the modelling of preferences and choice under bounded rationality.

A short list is a heuristic tool which is useful to DMs when certain information about offers can be obtained at low cost, but close inspection is required in order to obtain a fuller appraisal of the value of an offer. In general, heuristics should be well adapted to the cognitive abilities of decision makers, as well as the structure of the information obtained during the search procedure (see Simon [6,7], Todd and Gigerenzer [8], and Bobadilla-Suarez and Love [9]). Such lists are useful when exhaustive inspection of all the offers satisfying the fundamental requirements of a purchaser would be overly expensive in terms of search time or the amount of information needing to be processed exceeds the cognitive abilities of DMs (see Masatlioglu et al. [10], and Lleras et al. [11]). Borah and Kops [12] consider search processes where DMs construct short lists on the basis of information from peers. Short lists can also be useful when offers are categorised into different types (see Armouti-Hansen and Kops [13]).

Choice based on short lists is often implemented when offers are assessed according to various attributes. Kimya [14] considers a model of choice based on multiple attributes. The DM desires to choose an offer such that each attribute satisfies a given criterion. These criteria are considered in order of decreasing importance. As long as at least one offer under consideration satisfies the current criterion, offers that do not satisfy it are discarded. This procedure is followed until a single offer remains (this is the one chosen), all the criteria have been assessed or none of the offers satisfy the current criterion. In the final two cases, the DM chooses an offer at random from the set of offers presently under consideration. Such a procedure can be thought of as one based on short lists of successively smaller size.

When choice is based on a number of attributes, a number of approaches to decision making are available. One possible approach is called tallying. Using this approach, a DM chooses an offer which satisfies the most criteria. Such an approach can be adapted to take into account the importance of the criteria. Fechner et al. [15] consider the cognitive costs of various types of strategy for multi-criteria decision making.

The model presented here assumes that the attributes of an offer are split into two sets, denoted by $A_1$ and $A_2$. The attributes in set $A_1$ can be observed in an advert, while the attributes in set $A_2$ can only be observed by physically observing an offer. It is assumed that the costs involved in observing the attributes in set $A_1$ are much lower than the costs of observing the attributes in set $A_2$. For example, a DM searching for a flat can easily find adverts for offers that contain information regarding, for example, the size, location and price of a flat. However, in order to get a full appraisal of the attractiveness of an offer, the DM should physically observe an offer.

Of particular importance to the model considered here, Manzini and Mariotti [16] present a model of choice based on a number of rationales, denoted $(\mathcal{P}_1, \mathcal{P}_2, \ldots, \mathcal{P}_n)$. These rationales can be interpreted

as methods with which a DM compares a number of offers according to various sources of information. These comparisons are based on pairwise comparisons that are assumed to be asymmetric, i.e., if offer *a* is preferred to offer *b* under a given rationale $\mathcal{P}$, then offer *b* cannot be preferred to offer *a* under $\mathcal{P}$. Such a rationale may not be complete, for example, two offers may be incomparable or a DM might be indifferent between two offers based on the information he/she has. Also, such a rationale might not be transitive, that is, the following scenario is possible—offer *a* is preferred to offer *b*, which is preferred to offer *c*, but offer *c* is preferred to offer *a*. An offer is defined to be maximal under rationale $\mathcal{P}$ if and only if there is no offer preferred to it under $\mathcal{P}$. It is assumed that a decision maker sequentially removes all the offers that are not maximal under successive rationales until either only one offer is left or all of the rationales have been used. In the second case, the DM may choose an offer at random from the set of offers that have not been eliminated. Various properties of such procedures are considered, including when such procedures satisfy the so-called Weak Axiom of Revealed Preferences (WARP).

**Definition 1.** *A decision procedure satisfies WARP if the following condition is satisfied for any possible realisation of a search process under that procedure—when offers a and b belong to sets $S_1$ and $S_2$ and a is chosen from the set $S_1$, then b is never chosen from the set $S_2$.*

In the framework of this paper, a DM chooses an offer on the basis of two rationales, $(\mathcal{P}_1, \mathcal{P}_2)$. Rationale $\mathcal{P}_1$ will be interpreted as describing the preferences of a DM on the basis of initial information gained via the Internet and $P_2$ as describing his/her preferences on the basis of both this initial information and the information gained from physically observing an offer. The DM chooses which offers to physically view (the short list) on the basis of rationale $\mathcal{P}_1$ and then makes his/her final choice amongst the offers on the short list according to the rationale $\mathcal{P}_2$. The rationales considered are similar to those considered by Au and Kawai [17] who consider choice based on two rationales, $(\mathcal{P}_1, \mathcal{P}_2)$, where $\mathcal{P}_1$ is based on a transitive preference ordering and $\mathcal{P}_2$ is based on a complete and transitive preference ordering. This means that under $\mathcal{P}_2$ a DM can rank the offers under consideration according to a linear ordering. In the model presented here, both of the rationales satisfy this condition. On the basis of rationale $\mathcal{P}_1$, the DM chooses a subset $S_0$ of the set of offers $S$ which are all preferred to the offers in $S \setminus S_0$, that is, the DM forms a short list of the most highly-ranked offers on the basis of $\mathcal{P}_1$. Following this, the DM chooses the best offer from this short list on the basis of $\mathcal{P}_2$.

It should be noted that if offer *a* is preferred to offer *b* under rationale $\mathcal{P}_1$, then it is possible that offer *b* is preferred to offer *a* under rationale $\mathcal{P}_2$. In the context of the problem considered here, this means that although offer *a* is more attractive than offer *b* according to initial information, offer *a* is seen to be less attractive after close inspection of both offers. This means that such a procedure does not satisfy WARP, since it is possible to find two sets $S_1$ and $S_2$ such that *b* belongs to the short list and is then chosen when the DM chooses from $S_1$, but is not placed on the short list when choice is from $S_2$ and then option *a* is chosen.

It is assumed that in both stages of the search process, the offers can be assessed in parallel. However, in practical terms, this might mean that a set of pairwise comparisons are made in order to appropriately rank the offers. This will be considered in Section 2.2. Yildiz [18] considers a model of search in which offers are observed in sequence and either one or two rationales are employed. When a single rationale is used, the first offer to be observed is the initial candidate. Each successive offer is compared with the current candidate and replaces the current candidate when preferred to it. Once all the offers have been observed, then the current candidate is the one chosen. When two rationales are used, then all the offers that are not maximal according to the first rationale (which is possibly incomplete) are eliminated and then the second rationale is used to finally choose an offer. In fact, Yildiz [18] suggests a game of the form presented here in which one DM (the so-called Stackelberg leader) chooses a short list on the basis of his/her preferences and then a second DM makes the final decision of which offer to choose from this short list.

Similar decision making problems with multiple DMs have been been considered in an experimental or practical framework. For example, Giustinelli and Manski [19] investigate how families make practical decisions about the school a child should attend and Beynon et al. [20] consider how multiple DMs rank cars on the basis of multiple traits in an experimental setting. However, these decision problems are considered as processes of group decision making, rather than by using a game-theoretic approach. Here, it is assumed that under game-theoretic approaches DMs make decisions independently of each other and the outcome of a decision process depends on the individual decisions made. Under a group decision approach, each DM assesses (or ranks) the offers under consideration based on the current information. These assessments are then aggregated in order to make a decision. Alpern and Baston [21] and Sakaguchi and Mazalov [22] consider game-theoretic models of similar search problems to the one described here in the context of an employer looking for employees. However, these models consider sequential search without a two-stage inspection process. Horan [23] considers the properties of such two-stage search procedures based on transitive preference relationships.

This article considers the question of the length of short lists according to the parameters of a search problem, as well as the structure of the information. This is an obvious question which has so far received little attention in the literature. Another question regards adapting such models to situations where a decision process involves a number of decision makers.

The model presented here is somewhat similar to the one considered by Analytis et al. [24]. In the first phase (parallel search), a DM ranks offers based on initial information. In the second phase (sequential search), the DM closely inspects offers, starting with the highest ranked offer from the first phase, and stops searching (i.e., purchases an offer) when the offer's value exceeds the expected reward from future search. To carry out such a strategy, the DM must observe the values of offers, while in order to derive the optimal strategy, knowledge is required regarding the distribution of offers' values given the signal from in the first phase. The model presented here is adapted from the model considered by Ramsey [25] in the framework of a single DM using online and then offline search for a valuable resource. In the first phase, a fixed number, $n$, of offers are ranked on the basis of initial information. The $k$ most highly ranked offers from this initial inspection are then inspected closely. After this closer inspection, the highest ranked offer from the short list of offers is accepted. The signals observed in these two phases can be described by a pair of random variables $(X_1, X_2)$ from a continuous joint distribution. The DM cannot measure these signals precisely, but is able to rank offers on the basis of the signals observed. The DM's payoff is assumed to be given by a function of these signals minus the search costs that have been incurred. In order to realise such a strategy, the DM has to be able to rank offers according to the signals observed so far. In order to derive the optimal strategy, knowledge of the signal's joint distribution is required. This paper presents a game theoretic version of that search problem and considers in which scenarios such a search procedure can be practically applied.

Section 2 presents the original model of the search process with one DM. The problem of joint search by two DMs is considered in Section 3. Section 4 briefly describes a method for solving this game via the use of simulations. Some theoretical results are given in Section 5. A description of the numerical results are given in Section 6. Section 7 gives some conclusions and directions for future research.

## 2. Model of Short List Formation with One Decision Maker

### 2.1. General Formulation

This model was first presented by Ramsey [25]. A decision maker (DM) has to choose one of $n$ offers. Each of these offers are characterized by a pair of continuous random variables from a joint distribution. These variables may be interpreted as signals of the value of an offer. The DM first observes an initial signal of the value of each offer in parallel. The DM does not precisely measure the realization of this first variable, but is able to rank these signals according to a linear order, that is,

assign ranks to these initial signals from 1 (the best) to $n$ (the worst). This is called the initial ranking. A DM's strategy is defined by the length of the short list, $k$, where $1 \leq k \leq n$. When $1 < k < n$, then in the second round of inspection, the DM observes the second signal of the value of an offer for the $k$ best offers from the initial ranking. If $k = 1$, then the DM automatically chooses the best offer according to the initial ranking without observing the second signal. If $k = n$, then the DM observes all of the offers in both rounds of inspection. Assume that given the DM observes both signals for all the offers, then he/she can construct a linear ranking of these offers based on the combined signals. This will be called the overall ranking. Naturally, after the second round of inspection, the DM is only able to compare the $k$ offers on the short list with each other. This creates the DM's partial ranking. It is assumed that the partial ranking is consistent with the overall ranking, that is, offer $i$ is ranked above offer $j$ in the partial ranking if and only if $i$ is ranked above offer $j$ in the overall ranking.

In practice, such a procedure may be employed by an employer looking for a specialist employee. In this case, the first signal may well be a written application for the position from a candidate. The employer compares these applications and invites the candidates with the $k$ most highly ranked applications for interview. After interviewing these candidates, the employer offers the position to the candidate assessed to be the best on the basis of both the written application and the interview.

Denote the $j$-th signal corresponding to the $i$-th offer by $X_{i,j}$, where $1 \leq i \leq n$ and $j = 1, 2$. It is assumed that the pairs of signals $(X_{1,1}, X_{1,2}), \ldots, (X_{n,1}, X_{n,2})$ are independent and identically distributed realizations from the joint distribution with density function $f(x_1, x_2)$. Note that the signals associated with an offer may be correlated with each other, but the signals associated with different offers are independent of each other. The value of offer $i$, $V_i$, is a function of these two signals. It is assumed that $V_i$ is strictly increasing in both $X_{i,1}$ and $X_{i,2}$. In addition, it is assumed that when $x < y$ the distribution of $V_i$ given $X_{i,1} = x$ is stochastically dominated by the distribution of $V_i$ given $X_{i,1} = y$. It follows from this that $E(V_i | X_{i,1} = x)$ is strictly increasing in $x$.

Assume that the DM does not observe the values of these variables, but can make perfect pairwise comparisons between offers based on the signals observed. The goal of the DM is to maximize the expected reward from search, defined to be the value of the offer accepted minus the search costs. The search costs are split into the costs of initial inspection and costs of closer inspection. The costs of initial inspection, given by $c_1(k, n)$, are strictly increasing in both the number of offers and the length of the short list. These costs reflect the effort needed for initial inspection of the offers and control of the short list. The costs of closer inspection of the offers on the short list, given by $c_2(k)$, are assumed to be increasing in the length of the short list. Let $c(k, n) = c_1(k, n) + c_2(k)$ denote the total search costs. Intuitively, a short list of length $k$ should consist of the $k$ highest ranked offers from the initial ranking. This is due to the fact that the distribution of the reward obtained by choosing from these offers stochastically dominates the reward obtained by choosing from any other set of $k$ offers given the initial ranking.

### 2.2. Control of the Short List and Search Costs

It should be noted that it is not necessary to produce a full ranking of the offers based on the first signal, in order to construct a short list. It is assumed that the costs of constructing the short list are proportional to the number of pairwise comparisons that the DM carries out and the DM uses a two-step heuristic procedure which ensures that the number of pairwise comparisons carried out is close to the minimum required. It is assumed that the offers appear in random order in the first round of observations. A full ranking of the first $k$ offers is constructed on the basis of the optimal procedure for ranking a set of values using pairwise comparisons (described below). This creates an initial short list. In the second stage of the procedure, the DM first checks whether the present offer should be placed on the current short list. If not, then the DM immediately proceeds to the next offer. Otherwise, the present offer replaces the offer ranked $k$ on the current short list and is then ranked with respect to the other $k - 1$ offers on the current short list. After the initial signals have been observed for all of the offers, the current short list becomes the official short list.

First, we consider the procedure for ranking the initial $k$ offers. This ranking is constructed iteratively by ranking the $i$-th object to appear relative to the first $i - 1$ offers, for $i = 2, 3, \ldots, k$. Let $T_k$ be the expected number of pairwise comparisons needed to produce a full ranking of $k$ offers and $E_i$ the expected number of pairwise comparisons required to rank the $i$-th offer with respect to the first $i - 1$ offers. Hence, $T_k = \sum_{i=2}^{k} E_i$. Note that when $k = 2$, one pairwise comparison is needed, thus $E_2 = T_2 = 1$. Under the optimal procedure for ranking offers, the current offer is first compared with a median ranked item of the first $i - 1$ offers. Afterwards, the current offer is successively compared to a median offer from the subset of offers it needs to be compared with, until its position has been uniquely defined (see Knuth [26]).

For example, for odd values of $i$, the $i$-th item may be compared to the currently $\frac{i-1}{2}$-th ranked object (a median of the previous $i - 1$ offers). The $i$-th item is ranked more highly than this object with probability $\frac{i-1}{2i}$ and, in this case, it remains to compare this offer with $\frac{i-3}{2}$ others. Otherwise, it remains to compare this offer with $\frac{i-1}{2}$ others. Hence, for odd $i$

$$E_i = 1 + \frac{i-1}{2i} E_{(i-1)/2} + \frac{i+1}{2i} E_{(i+1)/2}. \tag{1}$$

When $i$ is even, regardless of whether the $i$-th item is better or worse than the median item of the $i - 1$ previous offers (ranked $\frac{i}{2}$), the procedure reduces to comparison with $\frac{i}{2} - 1$ previous offers. Hence, for even $i$

$$E_i = 1 + E_{i/2}. \tag{2}$$

From the $k + 1$-th offer onwards, control of the short list is carried out by comparing each new offer firstly with the offer presently ranked $k$ on the short list (this offer is labelled $D_k$). If the new offer is ranked higher than $D_k$, then the new offer replaces it and is ranked with respect to the other $k - 1$ offers presently on the short list based on the same approach as used in the first stage of the control procedure. Hence, for $i = k + 1, k + 2, \ldots, N$ at least one comparison is always made and with probability $k/i$, an average of $E_k$ additional comparisons are made. Hence, the expected number of comparisons made in the second stage (from offer $k + 1$ onwards) is given by $U_{k,n}$, where

$$U_{k,n} = n - k + \sum_{i=k+1}^{n} \frac{kE_k}{i}. \tag{3}$$

The total expected number of pairwise comparisons is $W_{k,n} = T_k + U_{k,n}$. Suppose that each pairwise comparison costs $c_a$, then the expected search costs in the first round of inspection are $c_a W_{k,n}$.

Note that for $i$ slightly greater than $k$, it might be more efficient to apply a procedure similar to the one used in the first stage. Using such a procedure, the DM should compare the present offer with the median ranked offer of the appropriate set of offers until it is decided that the present offer should not be placed on the current short list or occupies the appropriate position on the current short list. Such a procedure would lower the expected number of pairwise comparisons somewhat. However, this would come at the cost of making the procedure much less intuitive.

Once the short list has been finalized, the offers on it are then inspected more closely. It is assumed that after this round of inspection, the DM accepts the offer assessed to be the best overall. It should be noted that, after observing the first offer on the short list, it suffices to compare each successive offer with the currently highest ranked offer. It follows that $k - 1$ pairwise comparisons are required in the second round of inspection. Assuming the cost of each pairwise comparison in the second round of inspection is $c_b$, then the search costs incurred in the second round are $c_b(k - 1)$. Note that in practical problems of this type, the costs of gathering the information required to compare two offers may be much greater than the effort required then to decide which is the better of two offers. For example, if someone is searching for a new flat, the time required to travel to a flat and then observe it will be much greater than the time required to then mentally compare two flats. Hence, one might instead assume that the costs incurred in the second round of inspection are $kc_b$. However, modelling the

costs in this way simply adds $c_b$ to the overall costs, regardless of the length of the short list used. Hence, these two formulations of the costs in the second round lead to the same optimal strategy.

Note that due to the assumptions regarding the practical aspects of such search procedures, it is assumed that $c_b$ is much greater than $c_a$. For example, $c_b$ may be assumed to be the cost of inviting a specialist employee for interview. This includes travel expenses and the costs involved in using an interview panel. These costs will be much greater than the costs of comparing two written applications.

### 2.3. Specific Assumptions Regarding the Distribution of Signals

It is assumed that the pair of signals, $(X_1, X_2)$, describing the value of an offer come from a bivariate normal distribution. The signal $X_1$ is standardised, that is, it is assumed to come from a normal distribution with mean zero and standard deviation one. The mean of the second signal is also assumed to be zero and the coefficient of correlation between $X_1$ and $X_2$ is given by $\rho$. The residual variance of $X_2$, that is, the variance in $X_2$ that is not explained by $X_1$, is $\sigma^2$. Hence, given $X_1 = x$, then the marginal distribution of $X_2$ is normal with mean $\rho x$ and variance $\sigma^2$. The overall variance of the signal $X_2$ is thus $\frac{\sigma^2}{1-\rho^2}$. The value of an offer—$V$—is given by $V = X_1 + X_2$. From these assumptions, $E(V) = 0$ and

$$\text{Var}(V) = \text{Var}(X_1) + \text{Var}(X_2) + 2\rho\sqrt{\text{Var}(X_1)\text{Var}(X_2)} = 1 + \frac{\sigma^2}{1-\rho^2} + \frac{2\rho\sigma}{\sqrt{1-\rho^2}}. \tag{4}$$

It is simple to show that this variance is increasing in $\rho$. Note that $\sigma$ can be interpreted as the importance of the second signal relative to the first signal (taking into account the correlation between the two signals).

It should be noted that in sequential search problems where the values of offers come from a normal distribution, the expected duration of search under the optimal strategy depends on the ratio between the standard deviation of the values of the offers and the costs of observing an offer (see Real [27]).

## 3. A Game Theoretic Model of Short List Formation

Assume that two DMs, DM1 and DM2, together must choose one of $n$ offers using a procedure based on forming a short list. For convenience, DM1 will be referred to as "he" and DM2 as "she". One possible approach to this problem may be defined as follows—DM1 observes the first signal for each of the offers. On the basis of these signals, he selects a short list of offers to be inspected more closely. After closer inspection of the offers, DM2 then makes the final selection of the offer. Hence, DM1 can be treated as a Stackelberg leader. The strategy of Player 1 is his choice of the length of the short list.

When extending this model to one in which a pair of DMs are searching for a good that will be shared, the common interest between the DMs should be taken into account. This common interest might result from two sources: (a) the two DMs may have similar (correlated) preferences, (b) one DM may show good will (altruism) towards the other DM. It is assumed that DM1 shows good will towards DM2, while DM2 only considers her own payoff when making her decision. Note that this assumption will be considered in more detail in the conclusion. Based on this formulation, the optimal response of DM2 is trivial. She should observe the offers on the short list and choose the one that she ranks most highly. It is assumed that only DM1 incurs search costs in the first round of inspection (e.g., DM1 searches via the Internet), while both players incur search costs in the second round of inspection. The cost of a pairwise comparison in the first round of search is assumed to be 0.001. The cost of a pairwise comparison in round two is assumed to be 0.05. These values are chosen to (i) reflect the fact that inspection costs in the second round are much higher than in the first round, and (ii) to ensure that a moderate number of offers are placed on the short list under the optimal strategy when a single DM faces such a search problem. Due to the number of parameters included

in the game-theoretic model and the length of the paper, the aim of the paper is to investigate how the preferences of the DMs and the level of good-will between them affect the equilibrium. Hence, these search costs are fixed.

Under the model for a single DM, the two signals of the value of an offer may be correlated. One aspect of the common interest of DMs will lie in the fact that their appraisals of the value of offers may be correlated (i.e., the DMs may have similar preferences). In general, this might lead to a complex correlation structure between the signals observed by the DMs. In order to keep this correlation structure relatively simple, the following assumptions are made:

1. The correlation between the two signals observed by a single DM is $\rho_1$ (independently of the DM).
2. The correlation between the value of a signal as observed by the two DMs is $\rho_2$ (independently of the signal).
3. Given the value of the first signal as observed by a DM, the value of the second signal observed by this DM is conditionally independent of the value of the first signal as observed by the other DM.
4. Analogously, given the value of the second signal as observed by a DM, the value of the first signal observed by this DM is conditionally independent of the value of the second signal as observed by the other DM.

Let $X_{i,j}$ be the value of the $i$-th signal as observed by the $j$-th DM and let $\mathbf{X} = (X_{1,1}, X_{2,1}, X_{1,2}, X_{2,2})$ denote the set of signals of the value of an offer observed by the two DMs. The structure of the correlation between the signals is illustrated by Figure 1.

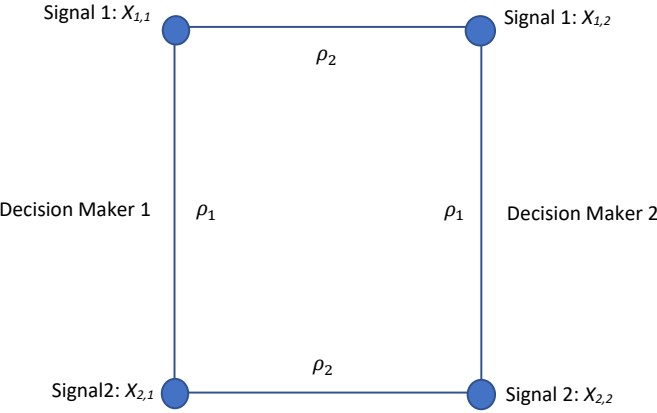

**Figure 1.** Structure of the correlations between the signals observed by the decision makers.

It follows that the correlation matrix for these signals is given by

$$\rho = \begin{pmatrix} 1 & \rho_1 & \rho_2 & \rho_1\rho_2 \\ \rho_1 & 1 & \rho_1\rho_2 & \rho_2 \\ \rho_2 & \rho_1\rho_2 & 1 & \rho_1 \\ \rho_1\rho_2 & \rho_2 & \rho_1 & 1 \end{pmatrix}. \tag{5}$$

The importance ascribed by the DMs to the two signals may differ. It is assumed that the importance of the second signal relative to the first signal for the $i$-th DM is given by $\sigma_i$. The standard deviation of the first signal as observed by DM1 is normalised to be equal to 1. The residual standard deviation of the second signal (given the value of the first signal) is assumed to be equal to $\sigma_1$. The value of an offer to DM1, $V_1$, is assumed to be given by $V_1 = X_{1,1} + X_{2,1}$. Analogously to the derivation of Equation (4), the overall variance of the value of an offer to DM1 is given by

$$\text{Var}(V_1) = 1 + \frac{\sigma_1^2}{1 - \rho_1^2} + \frac{2\rho_1\sigma_1}{\sqrt{1 - \rho_1^2}}. \tag{6}$$

Since the DMs might exhibit different levels of choosiness, $r$ denotes the relative choosiness of DM2 compared to DM1. It is assumed that the overall standard deviation of the value of an offer to DM2 is $r$ times the overall standard deviation of the value of an offer to DM2. It follows that $\text{Var}(V_2) = r^2\text{Var}(V_1)$. Since the covariance structure for the signals observed by DM2 is analogous to that of the signals observed by DM1, it follows that

$$\text{Var}(V_2) = k^2\left(1 + \frac{\sigma_2^2}{1 - \rho_1^2} + \frac{2\rho_1\sigma_2}{\sqrt{1 - \rho_1^2}}\right), \tag{7}$$

where the value of $k^2$ satisfies the equation $\text{Var}(V_2) = r^2\text{Var}(V_1)$. It follows that

$$k = \frac{r(1 - \rho_1^2 + \sigma_1^2 + 2\rho_1\sigma_1\sqrt{1 - \rho_1^2})}{1 - \rho_1^2 + \sigma_2^2 + 2\rho_1\sigma_2\sqrt{1 - \rho_1^2}}. \tag{8}$$

The interpretation of these parameters is important. Since, the standard deviation of the first signal observed by DM1 is standardised to be equal to one, the parameter $\sigma_1$ both describes the relative importance of the second signal to DM1 and his level of choosiness. However, $\sigma_2$ purely describes the importance of the second signal to DM2 relative to the first signal. As stated above, the parameter $r$ describes the level of choosiness of DM2 relative to DM1. It is assumed that the payoff obtained by a DM is the value of the offer accepted to the DM minus the search cost that he/she incurs.

In addition to having correlated assessments of the value of offers, DM1 may also show altruism towards to DM2, that is, the utility obtained by DM1 depends on the payoff of DM2. Let the level of altruism shown by DM1 to DM2 be measured by $\alpha$. Hence, the utility obtained by DM1 when he obtains a payoff of $y$ and DM2 obtains a payoff of $z$ is given by $u_1 = (1 - \alpha)y + \alpha z$ (see Markowska-Przybyla and Ramsey [28]). It is assumed that $0 \le \alpha \le 0.5$, where $\alpha = 0$ corresponds to DM1 being economically rational (i.e., DM1 only considers his own payoff) and $\alpha = 0.5$ corresponds to DM1 placing the same weight on the payoff of DM2 as on his own payoff. It is assumed that DM1 maximises his utility. The parameters used in the model are summarized in Table 1.

**Table 1.** Summary of the Parameters Used in the Game Theoretic Model.

| Parameter | Description |
|:---:|:---:|
| $\sigma_1$ | Relative importance of the second signal to DM1 |
| $\sigma_2$ | Relative importance of the second signal to DM2 |
| $\rho_1$ | Correlation between the two signals of the value of an offer observed by a single DM. |
| $\rho_2$ | Correlation between the values of a single signal as seen by the two DMs. |
| $\alpha$ | Level of altruism shown by DM1 to DM2. |
| $r$ | Level of choosiness of DM2 relative to DM1. |

Due to the mathematical difficulty of solving this game when the signals are correlated, various realisations of this game are solved with the aid of simulation. This involves simulating the values of the signals observed by the DMs. Firstly, a set of four signals, $\mathbf{Y} = (Y_{1,1}, Y_{2,1}, Y_{1,2}, Y_{2,2})$, whose covariance matrix is given by $\boldsymbol{\rho}$ is generated using the Cholesky decomposition (see Gentle

[29]). Using the Cholesky decomposition, the correlation matrix $\rho$ can be written as $\rho = \mathbf{L}\mathbf{L}^T$, where $T$ denotes transposition and

$$
\mathbf{L} = \begin{pmatrix}
1 & 0 & 0 & 0 \\
\rho_1 & \sqrt{1-\rho_1^2} & 0 & 0 \\
\rho_2 & 0 & \sqrt{1-\rho_2^2} & 0 \\
\rho_1\rho_2 & \rho_2\sqrt{1-\rho_1^2} & \rho_1\sqrt{1-\rho_2^2} & \sqrt{1-\rho_1^2-\rho_2^2+\rho_1^2\rho_2^2}
\end{pmatrix}.
\tag{9}
$$

Let $\mathbf{Y} = \mathbf{L}\mathbf{Z}$, where $\mathbf{Z}^T = (Z_1, Z_2, Z_3, Z_4)$ is a vector of independent realisations from the standard normal distribution, i.e., with mean and standard deviation equal to 0 and 1, respectively. The components of the vector $\mathbf{Y}$ have the appropriate correlation matrix, but the standard deviation of each component is equal to one. Hence, in order to obtain the components of $\mathbf{X}$, it is necessary to multiply these components by the appropriate standard deviations. Analogous to the model with one DM, the standard deviations of the signals observed by DM1 are 1 and $\frac{\sigma_1}{\sqrt{1-\rho_1^2}}$, respectively. The standard deviations of the signals observed by DM2 are $k$ and $\frac{k\sigma_2}{\sqrt{1-\rho_1^2}}$, respectively. It follows that

$$
X_{1,1} = Z_1; \quad X_{2,1} = \frac{\sigma_1 Z_2}{\sqrt{1-\rho_1^2}}; \quad X_{1,2} = kZ_3; \quad X_{2,2} = \frac{\sigma_2 k Z_4}{\sqrt{1-\rho_1^2}}.
\tag{10}
$$

## 4. Method of Empirically Solving the Game

The game described above is a type of Stackelberg game, where DM1 makes the first move and DM2 reacts (see Yildiz [18]). Such games are solved by recursion by first determining the optimal response of DM2 to the action of DM1. From the form of the game, this optimal response is trivial. DM2 should accept the offer on the short list which is most attractive to her (it is assumed that, when making their decisions, players do not have any information regarding the other DM's ranking of the offers). Given that DM2 uses such a strategy, DM1 should choose the length of the short list which maximizes his utility. It is intuitively clear that a short list of length $k$ should include the $k$ offers which are most attractive to DM1 according to the first signal (see Ramsey [25]). DM1 should choose $k$ in order to maximise his utility from the search procedure. In order to solve this game via simulations, the following algorithm may be used:

1. Simulate the values of the signals observed by the DMs for the $n$ offers $(X_{1,1}^i, X_{2,1}^i, X_{1,2}^i, X_{2,2}^i)$, for $i = 1, 2, \ldots, n$.
2. Form the appropriate short lists of length $k$ as chosen by DM1, for $k = 1, 2, \ldots, n$.
3. Determine which offer is chosen by DM2 for each of these short lists.
4. On the basis of this calculate the utility obtained by the DMs according to the length of the short list.
5. Repeat this process $N_{sim}$ times.
6. The equilibrium length of the short list (derived empirically) is the length $k^*$ which maximises the mean utility of DM1 from these simulations.
7. The estimated value of the game is the vector composed of the mean payoffs of the DMs from these simulations.

A more detailed description of these simulations is given in Section 6. Although the game is generally solved via simulation, for simple correlation structures the equilibrium can be derived without the need for simulations. Some theoretical results are given in the next section.

## 5. Theoretical Results

As the values of $\rho_2$ and $\alpha$ increase, the goals of the DMs become more coherent. In such a case, it is expected that the equilibrium behaviour exhibited will become more similar to the optimal behaviour of a single searcher. Some aspects of this relationship are described by the following two theorems:

**Theorem 1.** *Assume that $\rho_2 = \alpha = 0$. The equilibrium length of the short list is equal to one.*

**Proof of Theorem 1.** Under the assumption that $\rho_2 = \alpha = 0$, it follows that the expected utility of DM1 (here, his expected reward) from accepting an offer given his observation of the first signal of an offer is independent of all other factors. It follows that by including any other offer apart from the one which maximises his expected reward given the observation of the first signal can only decrease his expected reward from the search procedure. □

**Remark 1.** *The fact that DM1 should not allow DM2 any choice in such a game results from the complete lack of coherence in the preferences/goals of the players. In real life human decision making, it is expected that DM2 would not agree to such a procedure, since there are many possible mechanisms for making such a decision. Future research will look into such mechanisms and investigate the situations in which a given mechanism might be favoured and the robustness of such mechanisms.*

**Theorem 2.** *Assume that $\rho_2 = 1, \sigma_1 = \sigma_2$ and $r = 1$. The equilibrium length of the short list is equal to the equilibrium length of the short list for a single searcher where $\sigma = \sigma_1$, $\rho = \rho_1$ and DM1 faces the same search costs as in the game theoretic version.*

**Proof of Theorem 2.** Under these assumptions, the assessments of the DMs regarding the offers are identical and they have the same preferences. In this case, in the second round of search DM2 would always accept the same offer as DM1. Since in the Stackelberg game DM1 should choose the length of the short list to maximise his expected reward given the optimal reply of DM2, the equilibrium length of the short list is the same as in the corresponding problem where he is searching alone. □

It should be noted that when $\alpha = 0.5$, DM1 places as much importance on the payoff of DM2 as on his own. However, the individual preferences of the DMs may be completely different. In such a case, it is expected that the equilibrium length of the short list will be relatively large. The relation between the equilibrium length of the short list and the parameters will be investigated in the following section.

## 6. Simulations

A program was written in the R language to simulate such a search procedure. It is assumed that the total number of offers available is 100. In each of the problems considered, the cost for a pairwise comparison is 0.001 in round one of the search procedure and 0.05 in the second round. The number of parameter sets used was $3^6 = 729$, based on all the possible combinations of the following values of parameters:

$$r, \sigma_i \in \{0.5, 1, 2\}, i = 1, 2; \quad \rho_i \in \{0, 1/3, 2/3\}, i = 1, 2; \quad \alpha \in \{0, 0.25, 0.5\}.$$

For each of these combinations, 100,000 realisations of the search procedure were simulated in order to estimate the expected reward of the DMs under short lists of sizes between 1 and 20 (in each case the equilibrium length of the short list was less than 20).

### 6.1. Results from the Simulations

This section considers the effect of the parameters on the equilibrium length of the short list, the relative effectiveness of the search procedure (from the point of view of the sum of the utilities of the DMs) and the relative advantage of DM1 compared to DM2. For comparison, results for the corresponding search problems with one DM are also presented. It should be noted that the individual payoffs of the players are presented. In the case of DM1, these payoffs do not take the payoff of DM2 into account, that is, these are not the utilities of DM1 unless $\alpha = 0$.

### 6.1.1. The Optimal Strategy With one DM

Here, we consider nine realizations of the search problem described in Section 2 based on all of the combinations of the parameters $\sigma \in \{0.5, 1, 2\}$ and $\rho \in \{0, 1/3, 2/3\}$. The optimal lengths of the short list and expected reward from search are presented in Table 2. The first value in each cell is the expected reward from search and the second value is the empirically derived optimal length of the short list.

**Table 2.** Optimal rewards and optimal lengths of the short list with one DM based on 100,000 simulations (the first value in each cell gives the estimated optimal expected reward and the second the empirically derived optimal length of the short list). These results indicate that the length of the short list used by a single DM increases as the marginal value of the information from the second signal increases (i.e., as $\sigma$ increases and $\rho$ decreases).

|  | $\rho = 0$ | $\rho = 1/3$ | $\rho = 2/3$ |
|---|---|---|---|
| $\sigma = 0.5$ | (2.4611, 2) | (2.8953, 2) | (3.5542, 2) |
| $\sigma = 1$ | (2.8862, 5) | (3.6756, 4) | (4.9358, 4) |
| $\sigma = 2$ | (4.1905, 11) | (5.5895, 8) | (7.9163, 5) |

It can be seen that the optimal length of the short list is increasing in $\sigma$. Hence, search becomes more intense as the second signal becomes more important (it should be noted that as $\sigma$ increases the overall variance in the values of offers also increases). On the other hand, search becomes less intense as the two signals become more correlated with each other (keeping the residual variance of the second signal constant). It should be noted that although increasing $\rho$ also increases the overall variance in the values of offers, the first signal explains a greater proportion of the overall variance in the value of offers. This second factor is thus more important in determining the optimal length of a short list.

### 6.1.2. The Effect of the Level of Altruism from DM1

As mentioned previously, the parameter $\alpha$ does not affect the coherence of the preferences of the two players. However, as $\alpha$ increases, DM1 should take the preferences of DM2 into account more. As expected, as $\alpha$ increases, then the equilibrium length of the short list increases (all other things being equal). Firstly, we consider the results when $\alpha = 0$. When $\alpha = 0$ and $\rho_2 = 0$, then the equilibrium length of the short list is always equal to one, regardless of the values of the other parameters (see Theorem 1). When $\alpha = 0$ and $\rho_2 = 1/3$, then the equilibrium length of the short list is at most 2 and in a large majority of cases (74 out of 81) is equal to 1. The equilibrium length of the short list is only equal to 2 when the two signals observed by a DM are uncorrelated ($\rho_1 = 0$) and the second signal contains the majority of the information regarding the value of an offer to DM1 ($\sigma_1 = 2$). In such cases, the (limited) coherence of the preferences of the two players is sufficient for DM1 to give DM2 at least some choice. When $\alpha = 0$ and $\rho_2 = 2/3$, then the stronger coherence between the preferences of the DMs leads to DM1 giving DM2 a wider range of choice (a longer short list), especially when the second signal contains a large proportion of the information about the value of an offer.

As $\alpha$ increases the equilibrium length of the short list is generally greater, since DM1 takes the preferences of DM2 more into account. For a given level of altruism, the equilibrium length of the short list is clearly increasing in the choosiness of the DMs (i.e., in $\sigma_1$ and $r$). The largest equilibrium lengths of short list are obtained when $\alpha = 0.5, \rho_1 = 2/3, \rho_2 = 0, \sigma_1 = 2$ and $r = 2$ (either 15 or 16 in all three cases). In these cases, although the preferences of the DMs are independent, DM1 places the same importance on the payoff of DM2 as on his own. Hence, DM1 should allow DM2 a reasonably wide choice of offers to consider. Although the preferences of the DMs are not coherent, it is likely that an offer that is very attractive to both DMs can be found using a relatively long short list, due to the correlation between the signals observed by a single player.

**Table 3.** Effect of the level of good will shown by decision-maker (DM)1 on the efficiency and equilibrium length of the short list in games where $r = \sigma_1 = \sigma_2 = 1$ and $\rho_1 = 0$ (the first value in each cell gives the efficiency and the second the empirically derived equilibrium length of the short list). As the level of good will shown by DM1 increases, he gives DM2 a greater number of options to choose from. Also, when the preferences of the DMs are independent, the intensity of search by the DMs (i.e., the longest short list) is greatest when the correlation between the signals observed in the two rounds is high. In this case, close observation of a number of offers is very likely to lead to the selection of an offer which is attractive to both DMs.

|  | $\rho_2 = 0$ | $\rho_2 = 1/3$ | $\rho_2 = 2/3$ |
|---|---|---|---|
| $\alpha = 0$ | (0.4002, 1) | (0.5433, 1) | (0.7996, 2) |
| $\alpha = 0.25$ | (0.4895, 2) | (0.6457, 2) | (0.8360, 3) |
| $\alpha = 0.5$ | (0.5257, 4) | (0.6904, 5) | (0.8501, 6) |

When the DMs have the same level of choosiness, that is, $r = 1$, then we may define the relative efficiency of the search procedure in the game as the mean payoff of the DMs at equilibrium divided by the optimal payoff in the search problem with a single DM. Table 3 presents results illustrating the effect of the level of altruism on the equilibrium length of the short list and the relative efficiency of the search procedure in comparison to the optimisation problem faced by a single DM. It can be seen that the efficiency increases as the level of altruism of DM1 increases and the coherence between the preferences of the players ($\rho_2$) increases. When DM1 places as much weight on the payoff of DM2 as on his own, then the equilibrium length of the short list is similar to the optimal length in the search problem faced by a single DM, otherwise the equilibrium length of the short list is smaller than the optimal length in the problem faced by a single DM. It follows that when there is little "good will" between the two DMs, then splitting the roles of the players in the search process will cause a high level of conflict, unless their preferences are very coherent ($\rho_2$ is large).

### 6.1.3. The Effect of the Correlation Structure

Table 4 illustrates the effect of the correlation structure on the efficiency of search at equilibrium for various levels of altruism shown by DM1. For a fixed level of altruism, the level of correlation between to the two signals observed by a DM, $\rho_1$, has almost no effect on the efficiency. On the other hand, the level of coherence of the preferences of the players has a very strong positive effect on the efficiency. This is even stronger than the effect of the level of DM1's altruism (when $\rho_2$ is large, $\alpha$ only has a weak positive effect on the efficiency). The equilibrium length of the short list is generally increasing in the coherence of the DM's preferences, $\rho_2$, unless DM1 places the same weight on the payoffs of both DMs, $\alpha = 0.5$. When $\alpha = 0.5$, the equilibrium length of the short list is very similar to the optimal length in the corresponding search problem with one DM.

Figure 2 illustrates the relation between level of good will shown by DM1, the coherence of the preferences of the players and the effectiveness of the search procedure as modelled by the game in two cases where $r = \sigma_1 = \sigma_2 = 1$. In one case, the preferences of the DMs are independent, $\rho_1 = \rho_2 = 0$, and in the other case the preferences of the DMs are relatively coherent $\rho_1 = \rho_2 = 2/3$. It should be noted that the expected sum of the payoffs to the players given the length of the short list is independent of the level of altruism exhibited by DM1.

The sum of the payoffs is relatively less sensitive to the length of the short list when the preferences of the players are coherent. When the level of altruism exhibited by DM1 is equal to 0.5, then DM1 chooses the length of the short list in order to maximise the sum of the payoffs to the DMs. In both of the cases considered here, the corresponding equilibrium length of the short list is equal to four. This maximises the effectiveness of the search procedure given the parameters (it should be noted that this effectiveness will generally be less than 1, due to the incomplete coherence of the preferences of the DMs).

When DM1 does not show any altruism to DM2, then DM1 should simply maximise his own payoff. In both cases, this corresponds to DM1 choosing a short list of length one. From Figure 2, it can be seen that the level of altruism has a much greater effect on the effectiveness of search at equilibrium when the preferences of the DMs are independent. This is due to the fact that the corresponding sum of payoffs is a clearly lower proportion of the maximum sum than when preferences are coherent.

**Table 4.** Effect of the correlation structure on the effIciency and equilibrium length of the short list in games where $r = \sigma_1 = \sigma_2 = 1$ (the first value in each cell gives the efficiency and the second the empirically derived equilibrium length of the short list). The efficiency of the game-theoretic search procedure is increasing in the good will between the players and, in particular, the coherence of their preferences.

|  | $\rho_2 = 0$ | $\rho_2 = 1/3$ | $\rho_2 = 2/3$ |
|---|---|---|---|
| $\alpha = 0$ | | | |
| $\rho_1 = 0$ | (0.4002, 1) | (0.5433, 1) | (0.7996, 2) |
| $\rho_1 = 1/3$ | (0.4352, 1) | (0.5892, 1) | (0.8279, 2) |
| $\rho_1 = 2/3$ | (0.4608, 1) | (0.6220, 1) | (0.7817, 1) |
| $\alpha = 1/4$ | | | |
| $\rho_1 = 0$ | (0.4895, 2) | (0.6457, 2) | (0.8360, 3) |
| $\rho_1 = 1/3$ | (0.4355, 1) | (0.6735, 2) | (0.8515, 3) |
| $\rho_1 = 2/3$ | (0.4594, 1) | (0.6950, 2) | (0.8450, 2) |
| $\alpha = 1/2$ | | | |
| $\rho_1 = 0$ | (0.5257, 4) | (0.6904, 5) | (0.8501, 5) |
| $\rho_1 = 1/3$ | (0.5515, 5) | (0.7091, 4) | (0.8594, 4) |
| $\rho_1 = 2/3$ | (0.5750, 5) | (0.7238, 4) | (0.8658, 4) |

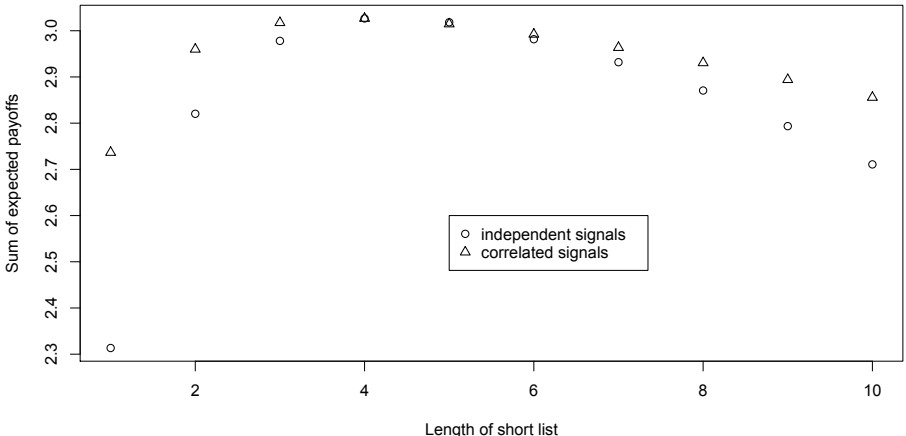

**Figure 2.** Effect of the length of the short list and correlation structure on the sum of the rewards obtained when $r = \sigma_1 = \sigma_2 = 1$. The correlated signals plot corresponds to $\rho_1 = \rho_2 = 2/3$ and the payoffs are scaled so that the maximum of the sum of the expected rewards are equal. This graph indicates that the efficiency of the game-theoretic procedure is much more sensitive to the good will between the DMs when their preferences are independent than when they are coherent.

### 6.1.4. The Effect of the Preferences of DM2

Tables 5 and 6 illustrate the effect of the preferences of DM2, described by $\sigma_2$ and $r$, on the payoffs at equilibrium and the equilibrium length of the short list. The parameter $\sigma_2$ describes the relative

importance of the second signal to DM2 and $r$ describes the overall choosiness of DM2 in comparison with DM1. Since DM1 chooses the length of the short list to maximise his utility, it is expected that DM1 will only give DM2 a choice of offers (choose a short list of size greater than one) when he exhibits altruism towards DM2 or the two DMs preferences are coherent.

**Table 5.** Effect of the preferences of DM2 on the payoffs and equilibrium length of the short list in games where $\sigma_1 = 1, \rho_1 = \rho_2 = 1/3$ (the first two values in each cell gives the payoffs of the players and the third gives the empirically derived equilibrium length of the short list). As the level of good will shown by DM1 increases, he is increasing willing to accept a decrease in his own expected payoff, in order to ensure an increase in the expected payoff of DM2. This is obtained by increasing the length of the short list, i.e., giving DM2 a wider range of offers to choose from.

| | $r = 1/2$ | $r = 1$ | $r = 2$ |
|---|---|---|---|
| $\alpha = 0$ | | | |
| $\sigma_2 = 1/2$ | (3.2457, 0.5925, 1) | (3.2498, 1.2493, 1) | (3.2476, 2.5475, 1) |
| $\sigma_2 = 1$ | (3.2472, 0.5138, 1) | (3.2466, 1.0845, 1) | (3.2415, 2.1933, 1) |
| $\sigma_2 = 2$ | (3.2483, 0.4053, 1) | (3.2425, 0.8667, 1) | (3.2510, 1.7970, 1) |
| $\alpha = 1/4$ | | | |
| $\sigma_2 = 1/2$ | (3.2431, 0.5971, 1) | (3.0509, 2.0209, 2) | (2.8692, 4.8467, 3) |
| $\sigma_2 = 1$ | (3.2491, 0.5209, 1) | (3.0824, 1.8684,2) | (2.7836, 5.0490, 4) |
| $\sigma_2 = 2$ | (3.2726, 0.3958, 1) | (3.1118, 1.6798, 2) | (2.8315, 4.7343, 4) |

**Table 6.** Effect of the preferences of DM2 on the payoffs and equilibrium length of the short list in games where $\sigma_1 = 1, \rho_1 = \rho_2 = 2/3$ (the first two values in each cell gives the payoffs of the players and the third gives the empirically derived equilibrium length of the short list). When DM1 shows no good will to DM2, then the preferences of DM2 have no effect on the length of the short list. The increase in the expected payoff of DM2 purely reflects the increased variance in the values of the offers to her. When DM1 shows good will, then the equilibrium length of the short list increases as the choosiness of DM2 increases.

| | $r = 1/2$ | $r = 1$ | $r = 2$ |
|---|---|---|---|
| $\alpha = 0$ | | | |
| $\sigma_2 = 1/2$ | (4.6080, 1.6407, 1) | (4.6058, 3.3453, 1) | (4.6030, 6.7091, 1) |
| $\sigma_2 = 1$ | (4.6046, 1.5361, 1) | (4.5992, 3.1176, 1) | (4.6039, 6.2983, 1) |
| $\sigma_2 = 2$ | (4.5977, 1.4030, 1) | (4.6021, 2.8571, 1) | (4.6066, 5.7818, 1) |
| $\alpha = 1/4$ | | | |
| $\sigma_2 = 1/2$ | (4.5968, 1.6442, 1) | (4.4526, 4.0190, 2) | (4.3002, 8.7310, 3) |
| $\sigma_2 = 1$ | (4.5028, 1.8762, 2) | (4.5013, 3.8403, 2) | (4.3680, 8.4348, 3) |
| $\sigma_2 = 2$ | (4.5400, 1.7738, 2) | (4.5352, 3.6442, 2) | (4.3004, 8.4916, 4) |

Tables 5 and 6 indicate that when DM1 does not show any altruism towards DM2 and given that $\rho_1 = \rho_2$, then the equilibrium length of the short list was found empirically to be equal to one. As the level of altruism shown by DM1 increases, then the equilibrium length of the short list increases, particularly when DM2 is relatively choosy and views the second signal as being more important than the first ($\sigma_2 = r = 2$). It can be seen that when DM1 exhibits altruism towards DM2, then he accepts a small decrease in his payoff in order to ensure a significant increase in her payoff. Hence, it may be said that the overall efficiency of the search procedure increases as the level of altruism exhibited by DM1 increases.

When DM1 shows no altruism towards DM2, then his payoff is independent of the preferences of DM2. On the other hand, when there is a positive correlation between the signals, the payoff of DM2 is decreasing in $\sigma_2$. This is true, since the equilibrium length of the short list is always one and the first signal gives less information about the value of an offer to DM2. When DM1 shows altruism to DM2, the relation between DM2's payoff at equilibrium and $\sigma_2$ is more complex and depends on the equilibrium length of the short list.

### 6.1.5. The Effect of the Relative Importance of the Second Signal to the DMs

Table 7 illustrates the effect of the relative importance of the second signal to the DMs on the effectiveness of the search procedure and the equilibrium length of the short list. When DM1 exhibits altruism to DM2, then the equilibrium length of the short list is clearly increasing in the relative importance of the second signal to DM1, $\sigma_1$, and weakly increasing in the importance of the second signal to DM2, $\sigma_2$ (it should be noted that this difference results from the fact that $\sigma_1$ is also positively related to the overall choosiness of DM1, while $\sigma_2$ only describes the relative importance of signal two to DM2). When DM1 does not exhibit altruism to DM2, then the efficiency of the search procedure is decreasing in both $\sigma_1$ and $\sigma_2$. This is due to the fact that the equilibrium length of the short lists is one in all cases and the amount of the variance in the value of an offer explained by the first signal is decreasing in the importance of the second signal to either DM. As the level of altruism exhibited by DM1 increases, the relationship between the efficiency of the search procedure and the relative importance of the second signal is less clear and depends on the equilibrium length of the short list.

**Table 7.** Effect of the relative importance of the second signal to the DMs on the effectiveness and equilibrium length of the short list in games where $r = 1, \rho_1 = \rho_2 = 1/3$ (the first value in each cell gives the effectiveness and the second gives the empirically derived equilibrium length of the short list). When DM1 does not show any good will, then the effectiveness of the game-theoretic solution is decreasing in the importance of the second signal, but this dependency disappears as the level of good will shown by DM1 increases. The effectiveness of the game-theoretic solution increases as the good will shown by DM1 increases.

|  | $\sigma_2 = 1/2$ | $\sigma_2 = 1$ | $\sigma_2 = 2$ |
|---|---|---|---|
| $\alpha = 0$ | | | |
| $\sigma_1 = 1/2$ | (0.6456, 1) | (0.6238, 1) | (0.5944, 1) |
| $\sigma_1 = 1$ | (0.6120,1) | (0.5892, 1) | (0.5590, 1) |
| $\sigma_1 = 2$ | (0.5466, 1) | (0.5236, 1) | (0.4924, 1) |
| $\alpha = 1/4$ | | | |
| $\sigma_1 = 1/2$ | (0.6459, 1) | (0.6930, 2) | (0.6700, 2) |
| $\sigma_1 = 1$ | (0.6899, 2) | (0.6735, 2) | (0.6518, 2) |
| $\sigma_1 = 2$ | (0.6741, 3) | (0.6655, 3) | (0.6688, 3) |
| $\alpha = 1/2$ | | | |
| $\sigma_1 = 1/2$ | (0.7197, 3) | (0.7084, 3) | (0.6886, 4) |
| $\sigma_1 = 1$ | (0.7166, 4) | (0.7091, 4) | (0.6943, 5) |
| $\sigma_1 = 2$ | (0.6983, 6) | (0.6977, 6) | (0.6871, 7) |

### 6.1.6. The Relative Advantage of Being the Stackelberg Leader and Practicality of such a Search Procedure

The expected payoffs of the players were compared in the cases where these payoffs were directly comparable ($r = 1$) and $\alpha < 0.5$ (i.e., DM1 placed a greater weight on his own payoff than on the payoff of DM2). For the 162 sets of parameters satisfying these conditions, the expected payoff of DM1 was greater than that of DM2 in 145 cases (89.51%). This indicates that the DM who chooses the initial short

list has an advantage in a wide range of scenarios. The follower (DM2) had the greatest payoffs in cases where (a) there was a relatively high degree of cohesion in the preferences of the DMs ($\rho_2$ large) and (b) the importance of the second signal to DM1 is large (either $\sigma_1 = 2$ or $\sigma_1 = 1$ and $\sigma_2 = 0.5$). In the majority of these cases (11 out of 17, 64.71%), DM1 also exhibited altruism towards DM2. In these cases, DM1 obtained relatively little information about the value of an offer from the first signal and due to the coherence of the preferences of the DMs, the assessment of the second signal by DM2 contains a significant amount of information about the value of an offer to DM1. For these reasons, in such a scenario DM1 should give DM2 a relatively large set of offers to choose from (i.e., the equilibrium length of the short list is relatively long). In such cases, intuitively it would probably be more effective to switch the order in which the DMs make decisions, that is, DM1 should choose the short list when the initial signal is of more importance to him and the second signal is of more importance to DM2.

On the other hand, when DM1 does not show any altruism to DM2 and the preferences of the DMs are not coherent ($\alpha = 0$ and $\rho_2 = 0$, respectively), then DM1 acts as a dictator by not giving DM2 any choice. This is due to the fact that the assessment of the second signal by DM2 contains no information about the value of an offer to DM1. This indicates a high level of conflict between the DMs.

Regarding the practicality of such a search procedure, it is expected that dividing the responsibilities of the DMs might be observed in practice when the efficiency of such a procedure is high. Although the efficiency of such procedures should be compared to other approaches to making a group decision, the following points can be made.

1. When DM1 shows no altruism to DM2 and the preferences of the players are incoherent, dividing the responsibilities of the DMs would only exaggerate the level of conflict in such a decision procedure. The Stackelberg leader maximises his reward by acting as a dictator. It is expected that the DMs would use a different approach to making group decisions in such scenarios.
2. The higher the level of goodwill between the DMs and the coherence of their preferences, the more practical it is to split responsibilities in the search procedure. It is suggested that when the preferences of the DMs differ, the leader should be the one who places relatively more importance on the first signal.

## 7. Discussion and Future Directions for Research

This article has presented a game theoretic model of two decision makers searching for a valuable resource that will be shared. According to this model, one of the decision makers acts as a Stackelberg leader by choosing a short list of promising offers based on initial information. The other decision maker then chooses an offer from the short list on the basis of additional information. This model takes into account the level of altruism showed by the leader to the follower, the coherence of the preferences of the decision makers, the correlation between the signals of the value of an offer as observed by a decision maker, the relative choosiness of the decision makers and the importance of the second signal relative to the first signal. The best response of the follower is to accept the best offer (according to her) on the short list on the basis of both the initial and additional information. The leader chooses the number of offers on the short list, $k$, which are to be inspected more closely. This number should be chosen to maximise the utility of the leader, which is taken to be a weighted average of the payoffs of the two decision makers. Hence, the equilibrium of such a game may be described simply by the length of the short list, $k^*$, which optimises the expected utility of the leader given that the follower accepts the offer on the short list that is most attractive to her. Such a model can be interpreted, for example, as a procedure for choosing a second-hand car on the basis of initial information gained via the Internet. One of the decision makers chooses a short list of offers to view on the basis of the technical specification of the car given in the advertisement. These offers are then inspected in real life and the second decision maker then chooses the car to purchase.

Some theoretical results have been given regarding this equilibrium. The relative advantage of being the leader and the effectiveness of such a search procedure were investigated on the basis of

simulations. When the players have the same level of choosiness, then the efficiency of the search process is defined to be the ratio between the mean reward of the payoff of the players at the equilibrium of the game and the optimal reward from search in the corresponding search problem faced by the leader. The efficiency of such a search procedure is increasing in the level of altruism exhibited by the leader and the coherence of the preferences of the players. When the leader does not exhibit any altruism to the follower and the preferences of the players are independent, then there is a large degree of conflict between the two players. The leader acts as a dictator and does not give the follower any choice. As the level of altruism exhibited by the leader increases, when the DMs do not differ in their choosiness, then the length of the short list at equilibrium increases towards the optimal length of the short list in the corresponding search problem involving just the leader. In cases where the leader exhibits a high level of altruism and the preferences of the players are independent, then this length of the short list at equilibrium can be even slightly longer than the optimal length of the short list when there is one searcher.

From the results presented above, it is expected that dividing the responsibilities between the players is practical only when the leader exhibits altruism towards the follower and/or the preferences of the players are relatively coherent. It is argued that when such a procedure is followed, then the player who places more importance on the first signal should take the part of the leader. For example, if the decision makers are looking for a second-hand car, then the individual who is more interested in the technical specifications of an offer (given in an advert) should play the role of the leader. The player who places more importance on the visual and tangible features of the offer should be the follower.

In future research, other game theoretic models where the final decision is constructed from decisions made by the individual decision makers could be compared to the model presented here. For example, in the first round of search the two decision makers could each choose a certain number of offers that should be inspected more closely. Each of the offers placed on the resulting short list is then inspected more closely. Some procedure is then used in order to choose the offer to be purchased. One can also develop models based on group decision making. Using such an approach, then each decision is jointly made by the decision makers on the basis of their appraisals of the attractiveness of the offers. For example, each player could individually rank the offers on the basis of initial information. These rankings can then be used to ascribe an initial score to each offer. A chosen number of offers with the highest scores are placed on a short list and inspected more closely. In the second round, each of the decision makers individually ranks the offers on the short list. These rankings may be used to define a final score for each of the offers on the short list. The offer with the largest final score is chosen. Such models should be compared in order to infer what procedures are most effective in given scenarios.

It should be noted that, in practice, the decisions made in both stages may be based on a number of criteria. In such cases, various methods for ranking based on multiple criteria (e.g., TOPSIS, see Yoon and Hwang [30], or MARS, see Górecka et al. [31]) can be used to rank offers according to the signals observed. The author intends to apply such an approach in the future. It should also be noted that when DMs are searching for offers using an Internet platform, the platform owner may have incentives to present information in a way that depends on the behaviour of a consumer (see Hefti [32]). Hence, game-theoretic models should be developed in which both platform users and platform owners are assumed to be DMs.

One somewhat paradoxical assumption of this model is that the Stackelberg leader (DM1) is assumed to show good will towards the follower (DM2), but that DM2 does not show any good will towards DM1. This paradox may be resolved by noticing that the good will shown by DM1 under such a search procedure is implicit in the size of the short list he chooses. Hence, DM1 does not need to have any information about how DM2 assesses the offers in order to show good will. On the other hand, in order for DM2 to show good will towards DM1 under such a search procedure, it is necessary for her to have information about how DM1 assesses the offer on the short list after closer inspection. Of course, in real life decision procedures of this form, DMs may exchange information

about their assessment of offers at each stage. One way of modelling such communication could be based on a group decision making approach. Another approach (particularly when there are more than two decision makers) could be based on the theory of cooperative games. Using such an approach, DMs would seek to achieve their goals by forming a coalition which is strong enough to make a choice that is beneficial to the group as a whole. Using such an approach, players can transfer utility between each other. For example, consider a DM who joins a coalition that then becomes strong enough to choose an offer which is favourable to the group as a whole, but not to the DM joining the coalition. The remaining members of the coalition could "recompensate" the joining member. The author intends to investigate approaches based on group decision making, as well as on the theory of cooperative games, in the future.

**Funding:** This research was funded by Polish National Science Centre grant number 2018/29/B/HS4/02857, "Logistics, Trade and Consumer Decisions in the Age of the Internet".

**Conflicts of Interest:** The author declares no conflict of interest.

## Abbreviations

The following abbreviations are used in this manuscript:

DM　　　Decision maker
TOPSIS　Technique for Order of Preference by Similarity to Ideal Solution
MARS　　Measuring Attractiveness near Reference Situations

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
