# Peer review of "A Game Theoretic Model of Choosing a Valuable Good via a Short List Heuristic"

_mathematics, doi:10.3390/math8020199_

Round 1

Reviewer 1 Report

The author presents a model of a game with two players when one player acts as a Stackelberg leader by choosing the short list of the goods based on some signal. Then, the second player acts as a  Stackelberg follower and chooses one good from the list suggested by the first player based on new additional signal. The utility of the first player takes into account the utility of the second player with some coefficient. So, this approach to define the utility represents a cooperative behaviour of the player. The paper contains two theoretical results presented in Theorems 1 and 2, in which some special cases of parameters of the model are considered and the equilibrium lengths of the short list created by Player 1 are obtained. As the explicit formulae for the equilibrium lengths are impossible to obtain in such a setting of the game, the author provides an extensive empirical analysis given in Section 6.  The calculations are made in R and represented in figures and tables.

 In my opinion, the model is very interesting and original. I have some questions and small remarks (typos) to the author:

Lines 215-216: Why are the search costs specified by parameters 0.001 and 0.05? Are these values motivated somehow or not? There is a suggestion for the future extension of the model. It is interesting to consider an approach when both players maximize the summarized utilities even when they act as Stackelberg players and have asymmetric information about the offers. After solving the maximization problem they can redistribute utilities using classical approaches from cooperative game theory or Nash bargaining solution or another bargaining solutions even if the utilities are non-transferable. Line 191: Double "it". Line 446: I think it should be "one in all cases..." instead of "one is all cases..."

In my opinion, the paper contains new and interesting research results, especially, if you take into account the applications of these results including the Internet search problem. I recommend the paper for publication in the journal "Mathematics".

Author Response

Thank you for your positive comments on my paper. In line with your comments, I have made the following changes:

1) I have explained the choice of the parameters defining the search costs. Since the goal of the paper is to look at the interaction between the two players and the number of parameters in the model, these parameters are assumed to be fixed.

2) In the conclusion, I have suggested that consideration of an approach based on cooperative game theory would be beneficial.

3) I have corrected the typos.

Reviewer 2 Report

In "A game theoretic model of choosing a valuable good via a short list heuristic" the author studies an interesting and vibrant problem with applications in the marketing, and in general helping us understand how and why we make certain decisions. The psychologists have long considered heuristics an internal decision-making mechanism we acquire over life on the basis of past decisions that seemed to have worked well.

While I have enjoyed reading this paper and find it contains promising results, I nevertheless have the following comments that should be taken into account if a revision will be granted at Mathematics.

1) I am little perplexed by the fact that author writes about group decisions, but then considers and studies a game with two players. In game theory, there is a clear distinction between games that are played pairwise (so between or involving two players), and games that are played in groups. The basic premise of this work seems to be contradictory to itself. The author should therefore carefully consider the words he uses for describing his approach.

2) In terms of group decisions being made, and game theory involving groups in general, there are fairly recently reviews available that might be useful to the readers, such as Evolutionary dynamics of group interactions on structured populations: A review, J. R. Soc. Interface 10, 20120997.

3) The introduction in general should be broader in scope, give the applicability of this research. I do not find the references give enough credit to related preceding research, and I find the focus on very old, albeit seminal, references rather counterproductive, and masking the actual novelty and contribution of this work.

4) It would also improve the paper if the figure and table captions would be made more self contained. In addition to briefly stating what is shown, one could add a sentence or two saying what is the main message of each table and figure.

5) Please double check references for completeness and correctness.

If a revision will be granted, I will be happy to review the manuscript again.

Author Response

Thank you for your constructive comments on my paper. In line with your recommendations, I have made the following changes:

1) I have made the differences between the concepts of game-theoretic and group decision approaches more apparent in the text. 

2) I have greatly extended the introduction to give a broader scope to the ideas presented in the paper. The bibliography has been appropriately extended to include a larger number of recent papers. The references [16]-[18] seem to be of much more specific interest to the ideas contained in the paper than the reference given by the reviewer.

3) Comments have been added to the tables and figures presenting the results.

Reviewer 3 Report

The paper presents a variation on the model in Ramsey (2019), allowing two decision makers to participate in the two-stage short list heuristic process. It allows signal importance, correlation of both signals and values, altruism, and choosiness to vary. Solutions are determined via simulation, and results are presented primarily as comparative statics. Some minor theoretical results are presented, but the clear results--and contributions--are that:

1) self-interest on the part of the first mover turns the problem into a trivial one: the short list generally consists of one option

2) altruism on the part of the first mover moves the problem toward that of a single decision maker receiving two signals

3) first-mover altruism renders relevant characteristics of the second mover that would otherwise have little influence on the outcome

4) the equilibrium length of the short list critically depends on the relative value agents place on the first-round and second-round signals, as well as coherence of preferences.

Overall, this is a good paper. The relevance to the literature is well-presented, the model is well-motivated and clearly described. The results are intuitive and the discussion is apropos.

Recommended changes:

A (very) brief gloss of the importance of costs: both the difference in c1 and c2 as well as the rationale for why DM1 bears both Are there productive thoughts on what would happen if DM2 also considered DM1's payoff? Again, a very brief digression, perhaps in the conclusion, might be worthwhile. Line 191: beginning has duplicative "it"

Author Response

Thank you for your positive comments on my paper. In line with your recommendations.

1) I have explained/considered the difference between the search costs incurred by the two players.

2) I have added comments in the conclusion on how DM2 might show good will to DM1 and how an approach based on cooperative game theory might be considered.

Round 2

Reviewer 2 Report

The authors have revised their manuscript comprehensively and with love to detail. I warmly recommend publication in present form.